# A Prototype to Detect the Alcohol Content of Beers Based on an Electronic Nose

**DOI:** 10.3390/s19112646

**Published:** 2019-06-11

**Authors:** Henike Guilherme Jordan Voss, José Jair Alves Mendes Júnior, Murilo Eduardo Farinelli, Sergio Luiz Stevan

**Affiliations:** 1Graduate Program in Applied Computing (PPGCA), State University of Ponta Grossa (UEPG), Ponta Grossa (PR) 84030-900, Brazil; henike_jordan@hotmail.com; 2Graduate Program in Electrical Engineering and Industrial Informatics (CPGEI), Federal University of Technology of Parana (UTFPR), Curitiba (PR) 80230-901, Brazil; mendes.junior13@yahoo.com.br; 3Graduate Program in Chemical Engineering, Federal University of Technology of Parana (UTFPR), Ponta Grossa (PR) 84016-210, Brazil; murilofarinelli@alunos.utfpr.edu.br; 4Graduate Program in Electrical Engineering (PPGEE), Federal University of Technology of Parana (UTFPR), Ponta Grossa (PR) 84016-210, Brazil

**Keywords:** electronic nose (e-nose), beers, alcoholic beverages, sensors, instrumentation

## Abstract

Due to the emergence of new microbreweries in the Brazilian market, there is a need to construct equipment to quickly and accurately identify the alcohol content in beverages, together with a reduced marketing cost. Towards this purpose, the electronic noses prove to be the most suitable equipment for this situation. In this work, a prototype was developed to detect the concentration of ethanol in a high spectrum of beers presents in the market. It was used cheap and easy-to-acquire 13 gas sensors made with a metal oxide semiconductor (MOS). Samples with 15 predetermined alcohol contents were used for the training and construction of the models. For validation, seven different commercial beverages were used. The correlation (R^2^) of 0.888 for the MLR (RMSE = 0.45) and the error of 5.47% for the ELM (RMSE = 0.33) demonstrate that the equipment can be an effective tool for detecting the levels of alcohol contained in beverages.

## 1. Introduction

Electronic noses (e-noses) are devices made from a matrix of chemical sensors based on metal-oxide materials. Like human noses, these devices are not able to identify substances separately in each sample. Data obtained by e-noses can be compared with a fingerprint because it is very difficult to find two different substances with the same pattern, then; it is possible to classify substances by these patterns [1]. E-noses are composed of four elements: a sensor matrix, signal processing unit, data storage, and pattern recognition. These four pieces simulate the data acquisition from the olfactory receptor neurons, the codification in the olfactory bulb, brain memory, and data processing performed by the human olfactory system, respectively [2]. 

Beverages have characteristics that distinguish them from each other and one the most important is their aroma. Commercially, the aroma has a fundamental role in attracting the consumer, besides being an indicator of product quality [3]. Several studies using e-noses have been developed to analyze the quality and characteristics of beverage as dairy products, coffees, fruit juices, and alcoholic beverages, being these last ones the most common [4]. 

There is a need for a commercial device which is portable and inexpensive, able to detect the volatile organic compounds (VOCs) present in alcoholic beverages (especially ethanol) and with high sensitivity sensors [5]. Therefore, the measurement of the alcohol emissions of certain a beverage (that is a VOC) is indicative of the quantity of this substance contained in that sample. There may be cases where certain beverages are labeled with alcohol values different from the real values, mainly due to the volatilization of this compound or other environmental factors. Besides that, studies show that in beers there are other VOCs such as acetals, esters, hydrocarbons, aldehydes, ketones, and carboxylic acids, which can be used as parameters to estimate the alcohol content. This makes it difficult to perform precise and instantaneous evaluations of these products. Thus, in line e-noses can be useful in the process of brewing of beverages, presenting themselves as an alternative to classical laboratory analyses. With the emergence of new microbreweries, is it essential to verifying the quality of the products during the beer production process. E-noses are suitable for this application as they are non-destructive and provide a quick and reliable response.

In the beverage industry, there are two techniques to analyze VOCs: gas chromatography with a mass spectrometer and quality analysis by sensory panels. These techniques are expensive and require considerable time (they may take up to a few days) [6]. The development of sensor technology has enabled e-noses to become simple devices with high accuracy, and these, in turn, are increasingly being used as an alternative to traditional methods. [7,8,9,10,11]. In addition, e-noses can also be applied to the monitoring of air quality [12,13,14,15,16,17,18], of gases emitted by the soil [1,19,20,21], in the evaluation of the food quality [22,23,24], the quality of wine [25,26], medical applications [27,28,29,30], among others.

This work presents the development of an e-nose prototype able to detect the alcohol content of beers. In this paper, the device is used for beers with alcohol contents between 1 and 8%. The device was trained with 15 different samples containing distillate water and alcohol (v/v), and then tested with seven commercial beverages. Four regression methods were applied to predict the alcohol content: neural network with a hidden layer (extreme learning machine), multiple linear regression, multiple nonlinear regression, and random forest. In all models, F test variance analysis was applied to identify if all the sensors contribute significantly to the model response.

## 2. Related Works

Nurul et al. [31] developed a method for rapid detection of ethanol concentration in beverages using an e-nose. It was tested beverages without alcohol, with 0.1%, 1%, and 10% alcohol concentration. In those results, the developed device could be used to quickly detect ethanol concentrations in several beverages, like alcoholic beverages, isotonic drinks, soft drinks, and fruit juices from different brands sold in Malaysia. The device demonstrated high precision and reliability, detecting ethanol concentrations as low as 0.1% (v/v). In addition, the authors used Response Surface Methodology to obtain a coefficient of determination (R^2^) of 0.9919, which means a high correlation between the model and the real response. 

Aleixandre et al. [32] build an e-nose for the quantification of wines. The wines were binary mixtures of two white wine and two red wine varieties. These beverages were elaborated by the traditional method using commercial yeasts. Partial Least Squares (PSL) and Artificial Neural Networks (ANN) methods were used to process the wine mixture measurements. For white wines, the R^2^ correlation between real and predicted values were 0.052 and 0.615 for PSL and ANN, respectively. For red wines, the results were 0.653 and 0.844. In this case, ANN showed better results for both wines. 

Reference [33] analyzed 21 different alcoholic beverages (beers, wines, and spirits) using an e-nose. Ragazzo-Sanchez et al. evaluated beverages after dehydration and dealcoholization procedures. Discriminant Factorial Analysis (DFA) and Principal Component Analysis (PCA) allowed them to clearly identify the differences among these beverages and classify them independently of the ethanol content. PCA showed better sample discrimination according to the ethanol content in aroma compounds, with a 97% clustering rate, achieving 97.9% for vodka, tequila, and whiskey, 88.1% for beer, and 87.4% for wines. Despite the good clustering results achieved by PCA, it is necessary to emphasize that the device did not discriminate the alcohol content but only among the beverages types and groups. 

Ghasemi-Varmkhast et al. [34] developed an e-nose based on MOS technology for beer aroma recognition. PCA was used as discrimination technique, showing a high separation between two groups: alcoholic beverages and non-alcoholic beverages. Reducing the components from five to two, the authors reported a variance close to 100% in the two principal components for their training and test datasets. Finally, a Support Vector Machine (SVM) was applied and the accuracy obtained for beer classification was 100% for both the training and test datasets. 

As described above, these papers focus on classification among beer groups and the development of e-nose systems that can avoid the procedures of dehydration and dealcoholization is an unexplored field. This is necessary because these procedures can result in loss and waste in beverage production (comparing label information and direct measurements of the beverages). This process can discriminate accurately between beverage samples, not only by their type but also by their alcoholic content. 

## 3. Materials and Methods

The methodology of this work is divided into four stages: e-nose construction, data acquisition from the experimental protocol, data preprocessing, and data analysis. Figure 1 shows the flowchart with each of these stages and the methodologies used.

### 3.1. Construction of Equipment

The selection of the sensors for the sensor matrix is the first stage in e-nose construction. Thirteen MOS gas sensors were used. Nine sensors (MQ-2, MQ-3, MQ-4, MQ-5, MQ-6, MQ-7, MQ-8, MQ-9, and MQ-135) belonged to the MQ line manufactured by Hanwei Electronics Co., Ltd. (Zhengzhou, China). Four sensors (TGS-822, TGS-2600, TGS-2602, and TGS-2603) were manufactured by TGS Figaro (Arlington Heights, USA). These sensors work as follows: when the target gas is present in the air, the conductivity of sensors changes, increasing as the gas concentration rises. This occurs due to the chemical reaction between the surface molecules in the semiconductor of the sensors and the gas molecules, which provides the change in sensor conductivity. The signals of each sensor are different due to the different semiconductor materials used. 

These 13 sensors were selected due to the fact they are inexpensive compared to other commercially available sensors and each sensor is sensitive to a different set of gases in addition to showing sensitivity to the target substance (alcohol). This number of sensors was chosen to create unique olfactory properties to generate a response for each gas (represented as a fingerprint). Therefore, the objective of this number of sensors is to analyze all the possibilities, identifying the sensors that have more contribution to discrimination of the target gas and, if necessary, to remove the redundant sensors. 

Table 1 shows the sensors used in this work and their most sensitive gases according to the corresponding Figaro^®^ and Hanwei^®^ datasheets.

Regarding the e-nose hardware, an ATMega 2560 was used due to its rapid prototyping. Since the microcontroller does not have all the analog inputs necessary to read all the sensors, a 16-channel Cd74HC4067analog multiplexer with one output was used. Thus, only one analog input was used. In addition, a pressure and temperature sensor (BMP180) and a humidity sensor (HIH-4939) were connected to the e-nose. These sensors are necessary because temperature, pressure, and humidity variations influence the responses of the gas sensors. One battery supplies the system.

Figure 2 illustrates the developed prototype, which was inserted into a box with 20 cm × 20 cm × 7 cm dimension. On the top, the box has holes in the cover and sides and a cooler to circulate air, which is pulled inside the device. The cooler was positioned above the sensors to provide a constant flow and homogenize the air reaching the sensors, while maintaining a stable temperature for measurements. In the center of the prototype, there is the array of sensors that contains the 13 gas sensors and below is the microcontroller. The device has a USB connection to a computer, which is responsible for data acquisition.

Figure 3 illustrates the e-nose block diagram. The process is summarized in the detection of the gas sample by the device, signal acquisition and conditioning circuits, 16-channel multiplexer, an analog-digital converter and serial data transmission from the microcontroller to the computer. 

### 3.2. Experimental Protocol

For e-nose calibration, 15 standard solutions with distilled water and ethanol (99.9%) were used (1%, 1.5%, 2%, 2.5%, 3%, 3.5%, 3.7%, 3.9%, 4.1%, 4.3%, 4.5%, 5%, 5.5%, 6%, and 8% v/v). Each solution had 100 mL volume. These values were chosen for being close to the concentrations of commercially available beers. These samples were used to train the database to find a model that adapts to any type of beer.

The standard solution was put into a beaker, positioned near the sensor matrix. The e-nose and the beaker were placed in a confined chamber with approximately 20 L volume aiming to create a homogeneous environment to avoid ethanol volatilizing from the solutions. After each measurement, the chamber was removed, cleaned, and the e-nose was reinstalled for a new measurement. Figure 4 shows this procedure. 

Six experiments were performed, two experiments by day. The first day was 25 °C and 33% of RH, the second day was 25 °C and 27%RH, and the third day 27 °C and 31%RH. Each experiment involved the measurement of 15 samples for 8 min and each sampling lasted 1 s. Thus, 480 data points were obtained for each sample (8 min × 60 s) and 2880 records (480 data points × 6 experiments) in total for each alcohol content. Finally, the training database had 43,200 records (2880 × 15 alcohol samples) and the algorithm models were built based on this database.

Each gas sensor requires a preheating period of about 10 min (according to the manufacturer) until they reach an optimum constant operating temperature. Therefore, these transient preheating data are discarded from the response analysis stage.

Seven types of beers of various brands commonly found on the market were used in the test stage. For each type of beer measurements of two samples of two different bottles/cans were made. Therefore, 14 beer samples (7 types × 2 samples) were measured for 8 min and each sampling lasted 1 s. Thus, 480 data points (8 min × 60 s) were obtained for each sample, and the test database had 6720 records (480 × 14 beer samples). The measurement in each test drink was performed at about 30 °C and 41%UR. For all experiments, the sample rate of data acquisition was 1 sample/sec.

No previous physical-chemical analysis of those beverages was performed, therefore, for these samples, the used as reference values those labeled on the bottles. According to Normative Rule N°54, dated 5 November 2001, page 5, which adopts the MERCOSUL technical regulation on brewery products, it is stated that the declaration of the alcohol content (except for non-alcoholic beer and malt) is expressed as a percentage by volume (% vol.) with a tolerance of ±0.5 vol. [35]. Therefore, breweries should ensure that beverages have a maximum difference of 0.5% of alcohol above that contained in the label, and this information was adopted as the basis for the study of this work. Table 2 summarizes these beers used in this paper with their labeled alcohol contents and licensed alcohol content range.

### 3.3. Preprocessing

A modified moving average filter was used to preprocess the signal. This filter was used because it is simplest, fastest, robust, and is easy to implement. Besides that, this filter obtained a satisfactory response compared to the filters mentioned in [36], such as exponential moving average, integral response, and differential response. A simple moving average over n elements consists of the unweighted averages of the subsets of n elements in a dataset, being n adopted as 50. The modification made was that only the values of the mean moving with the maximum difference of three times the standard deviation enter the moving average equation, i.e., |p_i_ − μ|≤ 3σ. Therefore, the value of n is 1 ≤ n ≤ 50 and within the normal distribution curve, 99.74% of data will be around the mean plus three standard deviations. This change provides more smoothing in the dataset, removing possible signal peaks due to the noise coming from the circuit.

The gas sensors have the behavior that their response varies according to the temperature and the relative humidity of the air, which the manufacturers’ datasheets show as characteristic curves of the sensors with the variation of these two variables. These curves use the resistance of the sensors at room temperature and clean air (R_0_) and varying conditions (R_s_). Load resistance (R_L_) changes the sensitivity of the sensors, and in this paper, all the load resistors were set to 10 kΩ. Therefore, since temperature, humidity, and pressure were not controlled, this adjustment is necessary to compensate for variations.

Using the data from the curves reported by the manufacturers, response compensation was performed as a function y(R_s_,x_0_,x_1_), where R_s_ is the measured resistance, x_0_ is the measured temperature, and x_1_ is the relative humidity of the air. The equations for all sensors have the form shown in Equation (1):(1)$$y=\frac{R_{s}}{\alpha_{0}x_{0}+\;\alpha_{1}x_{0}{}^{2}+\beta_{0}x_{1}+\;\beta_{1}x_{1}{}^{2}+\;\delta x_{0}x_{1}+\;\xi }$$

Thus, the difference between the sensors is in the constants $\alpha_{0}$, $\alpha_{1}$, $\beta_{0}$, $\beta_{1}$, δ e ξ. Table 3 presents these values for the 13 sensors besides R^2^.

Thereon, the data normalization (p_inorm_) was performed, which is given by Equation (2):(2)$$p_{inorm}=\;\frac{p_{i}-min\left(p\right)}{max\left(p\right)-min\left(p\right)}$$ where min is the minimum value and max is the maximum value of sample i from p set.

### 3.4. Data Analysis

According to the literature, it is more common to find predictive models applying regression and classification. The fundamental difference between these two types of models is that the classification can label the outputs through discrete classes, that is, it requires the samples to be classified into one of two or more predefined classes. On the other hand, models that apply regression can predict problems in a continuous quantity, that is when the problem requires quantity forecasting. Analyzing the problem of the alcohol content detection, we see the need for models capable of delivering the response as a value of the amount of alcohol in each sample; and since this sample can have values that vary continuously between 1% and 8%, the models that best fit this problem are those that apply regression. The application of classification models, in the case of alcohol detection, would require a training base with a larger spectrum of labeled classes, since the classification application for prediction of continuous values provides this value in the form of a probability for a label of class. In addition, regression allows predicting quantitative values with one, two or more decimal places of precision, which is of paramount importance in such applications.

Four regression methods were used: Multiple Linear Regression (MLR), Multiple Non-Linear Regression (MNLR), Random Forest (RF), and Extreme Learning Machine (ELM), a type of ANN.

MLR can be used in a more realistic scenario of dependency of several variables. This technique is like simple linear regression but the dependent variable (y) depends on two or more input variables, not just one (x). Regarding non-linear models, their main advantages are the ability to interpret and predict [37]. The difference between MLR and MNLR is that the latter that has as characteristic non-linearity, which includes polynomial, exponential, logarithmic models, among others.

ELM is single-hidden-layer feedforward neural network whose learning speed can be thousands of times faster than traditional network learning algorithms such as the backpropagation algorithm, obtaining a better generalization performance. Unlike traditional learning algorithms, the ELM learning algorithm not only tends to achieve the smallest training error but also lower weights [38]. This ANN was used because the training time is very short, and the online version allows updating the model using a small piece of the training set in each iteration, being hidden layer size and the learning function the only parameters to be adjusted.

Random Forest (RF) is a group classifier that produces multiple decision trees using a subset of samples and training variables randomly selected. This algorithm is used both for classification as regression [39]. For RF is not necessary to perform the pruning method of some nodes to avoid overfitting. Some recent papers show that the use of this technique in e-noses can improve the accuracy of regression/classification [40,41]. 

K-fold cross-validation was used to validate the obtained results for all four methods. This technique is a commonly used and takes a set of m examples and partitions them into k sets of equal sizes (called folds) of size m/K. For each set, a model is trained in the other sets [42].

An F-test was applied to analyze the variance with a confidence interval of 95% to verify if one or more variables (from gas sensors) cannot significantly contribute to the proposed model. For this regression model, the t-test is used to determinate if has a linear relationship between the response (Y) and some regressive variables (x_0_,…,x_n_). 

All algorithms were implemented in R software and Java, which have several libraries and functions that aid in the model building process and the real-time data acquisition system. For data storage, a database was built using PostgreSQL.

## 4. Results and Discussion

Figure 5a–d presents the typical sensor responses (non-normalized) to all calibration samples of 1%(v/v), 3.9%(v/v), 6%(v/v), and 8% (v/v), respectively, produced with ethanol and distilled water. The measurement time was about 480 s.

From Figure 5, it can be noted that the response of the sensors tends to grow with an increasing amount of alcohol in the sample, especially the MQ-3 sensor. This MQ-3 (the sensor most sensitive to alcohol) was the sensor that increased its voltage response from approximately 0.75 V in a transient state for the sample from 1% to approximately 2 V in the sample 3.9%, 2.5 V in the sample of 6% and finally, about 3 V for the 8% sample. The high sensitivity of the sensors and the assurance of repeatability of these responses, repeating the calibration procedure with the standard samples, indicate how reliable the equipment can be.

The statistical F-test verified that the MQ-135 (*p*-value of 0.0949) and TGS-822 (*p*-value of 0.3725) sensors can be removed from the input of regression models. This decision was made because the *p*-value was greater than the significance index. The H_0_ hypothesis is accepted, therefore, these explanatory variables do not contribute significantly to the construction of the models. 

Table 4 shows the *p*-values for each gas sensors used in this study. Thus, 11 gas sensors were used as input to the four regression methods using the calibration data. For MNLR, a logarithmic transformation was used because it was noted that the data tended to stabilize to high alcohol contents. However, the correlation was greater for MLR, which suggests that for the reduced samples (up to 8% of alcohol) the dataset tends to be linear. Regarding EML, with a reduced number of neurons in the hidden layer, the model responded with high error rate. With a large hidden layer, the model did not present significant changes in error rate, which indicates that the model had overfitting. For this application, 15 neurons in the hidden layer gave the best results. The activation function chosen was purely linear because it presented better precision in the cross-validation step of the dataset. For RF, the ideal number for this application was 500 and the number of variables by level was four. In the validation step, these parameters represent the best results from the training dataset. 

Table 5 presents the parameters obtained for each method, where R^2^ is the coefficient of determination (which indicates how much the model can explain the observed values), R^2^ adj is the adjusted R^2^ coefficient (which is the percentage of variation in response that is explained by the model adjusted for the number of predictor of the model in relation to the number of observations), *nhid* is the number of neurons in the hidden layer of EML, *actfun* is the activation function used in neurons, *ntree* is the number of trees, and *mtry* is the number of variables randomly sampled as candidates in each division in the RF method. 

The results show that the RF presented the worst performance, with difficulty of predicting values between 4.5% and 5%. This is important due to the fact many beverages have alcohol contents in this range. Except for these three samples, RF presented a reduced error rate (less than 3% of alcohol) within the tolerance range. C4M (a dark beer) sample had a higher response for all the algorithms compared to the C4 (beer) sample. Although both have 4%, C4M has a characteristic odor (stronger and more striking than C4) because it has ingredients are capable of emitting more VOCs.

The results obtained by the EML for all the samples (except C5.4) gave an alcohol difference less than or equal to 0.5%, which is the tolerance range. In addition, ELM obtained the closest labeled value for all samples, with a difference of 0.01 for C7.9. Considering the tolerance of ±0.5% imposed by the normative rules, MLR and MNLR obtained results within the range of acceptable alcoholic graduation from C4 to C5, having some difference in C5.4 and C7.9 of one decimal place. Therefore, ELM obtained the most consistent results considering the normative standard. Figure 6 presents the predicted values of the alcohol content for all the techniques used in this paper (MLR, ELM, RF, and MNLR).

Table 6 presents the mean squared error (RMSE) for each applied method in both validations (k-fold with k = 10) and test steps for calibration samples. 

From Table 6 and Table 7, it can be concluded that the methods with the best prediction rate were ELM followed by MLR. Both presented an average error rate of less than 8% and a mean square error of less than 0.5, demonstrating that for most of the tested beers, the error was in one decimal place. MLR had a close fit for C4M and EML for C7.9. Both were very close to C4.5. The ELM obtained the best performance in comparison to all the algorithms used in this paper. Also, noteworthy is the considerable error rate for C5.4 in all applied methods (above 7%). Table 7 shows the percentage of error between the predicted value and the labeled value of the beers, in addition to the average error percentage for each method applied.

Table 8 summarizes the results obtained from related works, along with the techniques used and their parameters. It is noteworthy that [34] built a device with a prediction rate of 100% to differentiate whether there is alcohol in beers or not, therefore, the device did not stipulate the percentage in certain samples. Reference [31] tested the equipment with three samples of alcohol and distilled water −0.1%, 1%, and 10% (v/v). Despite having achieved a high correlation, the performance of the device has not been tested with actual samples of beers or distillates. 

Compared with other works found in the literature, the R^2^ of the model constructed in this work was better adjusted than all models in [32], which demonstrates the good fit found here. In addition, the RMSE found for ELM and MLR, of 0.33 and 0.45, respectively, were within the tolerance margin imposed by normative instruction [35], which is 0.5.

## 5. Conclusions

To develop a device capable of detecting the alcohol content of some types of beers, a cooling system based on ambient air was successful in the alcoholic content prediction of various types of beers. The seven samples with a different alcohol content that were used to train and build the models proved to be sufficient and an appropriate methodology to detect the ethanol levels in beers.

The device showed reliability because it could detect volatile compounds emitted by beverages and indicated the amount of alcohol with high sensitivity. The percentage differences are less than 0.5% for some types of commercial beers. RMSE was less than 0.5 for the validation and test samples. In addition, the percentage error was less than 5.5% for ELM in beer samples, showing that this device reached a satisfactory accuracy in the prediction of alcohol content in relation to the other methods, considering the percentage error. The MLR showed a higher correlation than the MLNR, so the model had more linearized characteristics, besides being the most consistent method with the reality considering a tolerance of ±0.5%. RF had the worst performance compared to the others, in both the mean hit rates and the validation and test RMSE. Therefore, RNA with regression adjustment/classification capability, adaptable to linearized and no-linearized models, was able to overcome the simplicity of linear regression, thus being an algorithm with actions closer to expected. 

It was possible to remove the MQ-135 and TGS-822 sensors from the dataset, without prejudice to model prediction which allows for cost reduction, device size, power consumption, and autonomy. It is worth noting that the device is generic and can adapt to a variety of applications. In this application, the sensors MQ-135 and TGS-822 were redundant or not relevant, but could be more significant in other applications and therefore were not removed from the prototype.

As future works it is desired to train the device with samples more compatible with the beverages to be tested, since the beers do not only release alcohol as a volatile compound, minimizing the error of prediction of unknown samples. The major compound in beers is ethanol, but other volatile compounds such as esters, hydrocarbons, acetals, aldehydes, ketones, carboxylic acids are identified in those beverages. Therefore, the device can be used to classify and differentiate types of beverages with close alcohol levels, so this is a potential application. In addition, applying new tools and algorithms to have a better precision to identify in beer samples and to prove the alcohol content in each sample with the physical-chemical analysis to obtain a greater characterization of these beverages. 

## Figures and Tables

**Figure 1 sensors-19-02646-f001:**
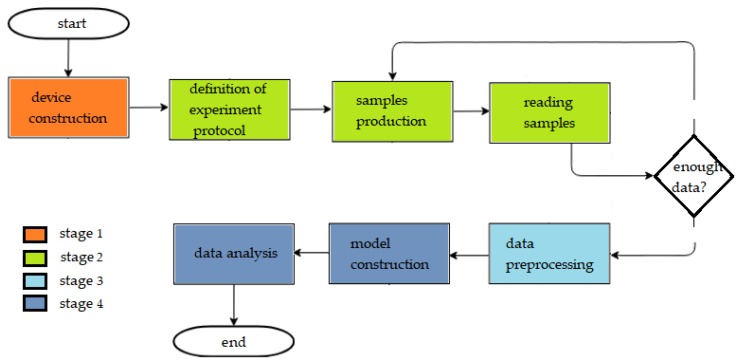
Flowchart of the adopted methodology.

**Figure 2 sensors-19-02646-f002:**
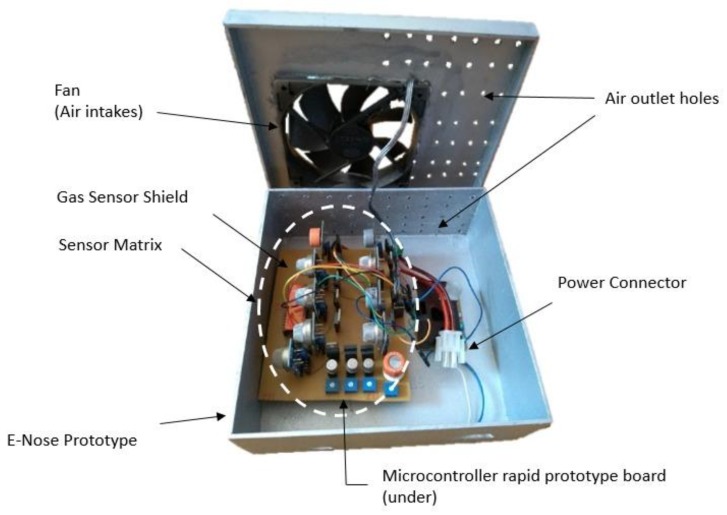
Equipment developed.

**Figure 3 sensors-19-02646-f003:**
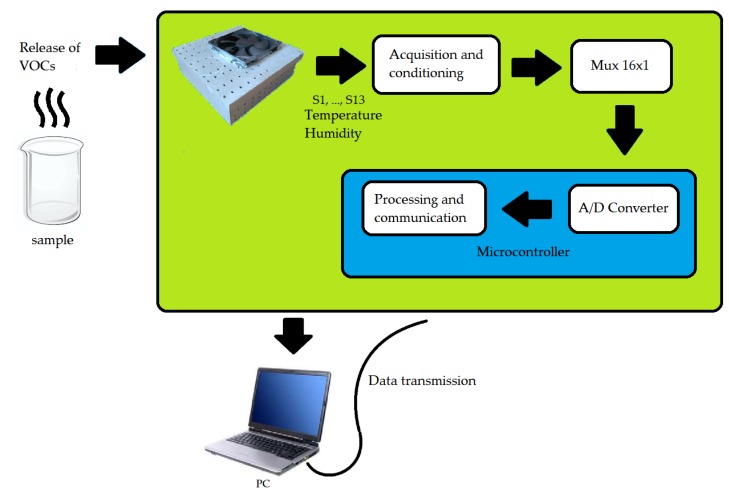
Circuit block diagram.

**Figure 4 sensors-19-02646-f004:**
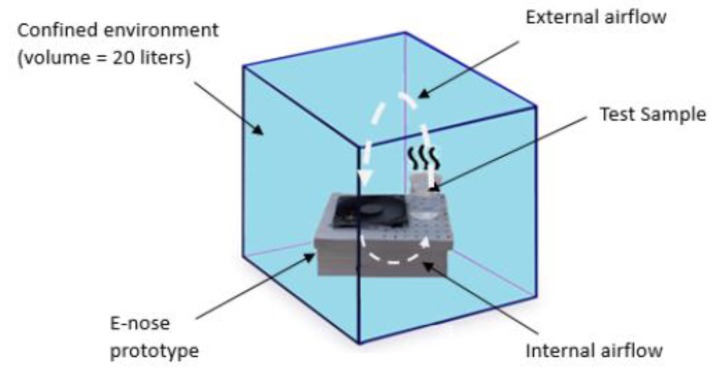
Standard solution sampling step.

**Figure 5 sensors-19-02646-f005:**
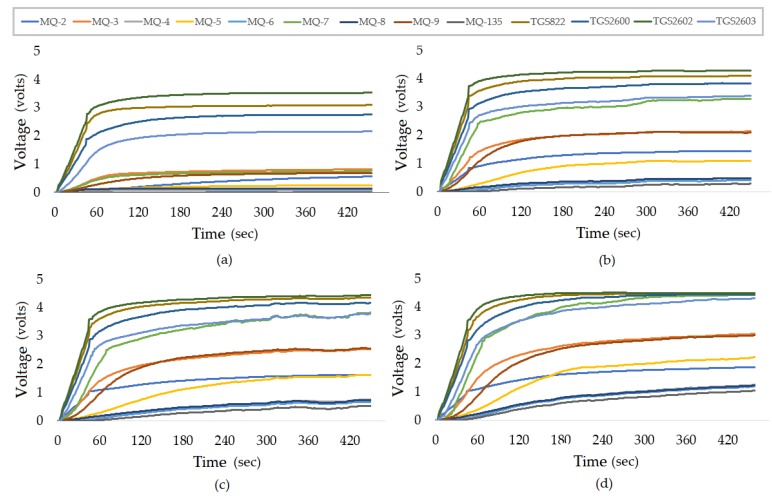
Sensor response. (**a**) for the 1% sample; (**b**) for the 3.9% sample; (**c**) for the 6% sample; (**d**) for the 8% sample.

**Figure 6 sensors-19-02646-f006:**
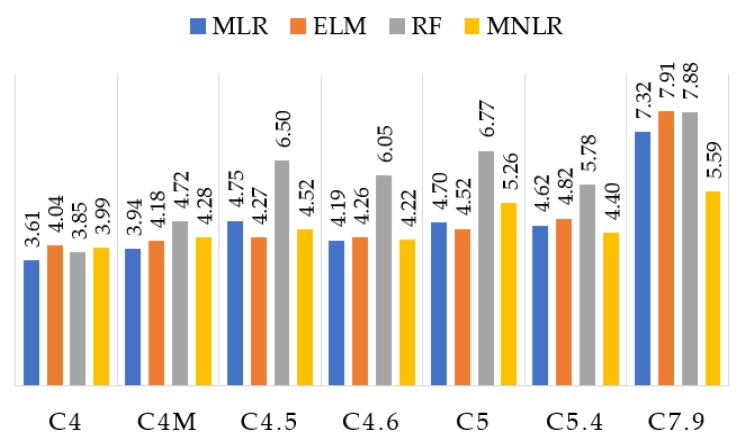
Predicted values of alcohol content (%) for each of the methods applied.

**Table 1 sensors-19-02646-t001:** Sensors used in the experiments and their sensitive gases.

Sensor	Sensitive Gases
MQ-2	$H_{2}$, LPG, ${\mathrm{CH}}_{4}$, CO, **ethanol**, propane, butane, and methane
MQ-3	**Ethanol**, benzene, ${\mathrm{CH}}_{4}$, hexane, LPG, and CO
MQ-4	LPG ${\mathrm{CH}}_{4}$, $H_{2}$, CO, and **ethanol**
MQ-5	$H_{2}$, LPG, ${\mathrm{CH}}_{4}$, CO, **ethanol**, isobutane, and propane
MQ-6	LPG, $H_{2}$, ${\mathrm{CH}}_{4}$, CO, **ethanol**, isobutane, and propane
MQ-7	CO, $H_{2}$, LPG, ${\mathrm{CH}}_{4}$ and **ethanol**
MQ-8	$H_{2}$, LPG, ${\mathrm{CH}}_{4}$, CO and **ethanol**
MQ-9	CO, ${\mathrm{CH}}_{4}$, and LPG
MQ-135	${\mathrm{NH}}_{3}$, benzene, **ethanol**, ${\mathrm{CO}}_{2}$, CO, and ${\mathrm{NH}}_{4}$
TGS822	Acetone, n-hexane, benzene, **ethanol**, isobutane, CO, and methane
TGS2600	$H_{2}$, CO, methane, isobutane, and **ethanol**
TGS2602	$H_{2}$, ${\mathrm{NH}}_{3}$, **ethanol**, $H_{2}S$, and toluene
TGS2603	$H_{2}$, $H_{2}S$, **ethanol**, methyl mercaptan, and trimethylamine

**Table 2 sensors-19-02646-t002:** Commercial beverages used to test the models.

Name	Description	Labeled Alcohol Content	Alcohol Content Licensed Range
C4	Pure Malt Beer	4%	3.5–4.5%
C4M	Malzbier Beer	4%	3.5–4.5%
C4.5	Pilsen Beer	4.5%	4–5%
C4.6	Pure Malt Beer I	4.6%	4.1–5.1%
C5	Pure Malt Beer II	5%	4.5–5.5%
C5.4	Black Beer Stout	5.4%	4.9–5.9%
C7.9	Mixed Beer	7.9%	7.4–8.4%

**Table 3 sensors-19-02646-t003:** Values of the constants used in the adjustment equations of the sensors.

Sensor	$\alpha_{0}$	$\alpha_{1}$	$\beta_{0}$	$\beta_{1}$	δ	ξ	R^2^
MQ-2	−0.0266	0.0003	−0.0023	0.0000	0.0000	1.4700	0.9989
MQ-3	−0.0229	0.0002	−0.0036	0.0000	0.0000	1.4640	0.9960
MQ-4	−0.0103	0.0001	−0.0033	0.0000	0.0000	1.2760	0.9970
MQ-5	−0.0145	0.0001	−0.0040	0.0000	0.0000	1.3350	0.9881
MQ-6	−0.0126	0.0001	−0.0034	0.0000	0.0000	1.2920	0.9941
MQ-7	−0.0157	0.0001	−0.0036	0.0000	0.0000	1.3580	0.9876
MQ-8	−0.0106	0.0001	−0.0012	0.0000	0.0000	1.0880	0.9962
MQ-9	−0.0158	0.0001	−0.0037	0.0000	0.0000	1.3590	0.9875
MQ-135	−0.0258	0.0003	−0.0023	0.0000	0.0000	1.4660	0.9978
TGS822	−0.0576	0.0005	−0.0179	0.0001	0.0001	2.7390	0.9910
TGS2600	−0.0825	0.0008	−0.0187	0.0000	0.0003	3.1140	0.9907
TGS2602	−0.0566	0.0004	−0.0070	0.0000	0.0001	2.2480	0.9944
TGS2603	−0.0400	0.0002	0.0000	0.0000	0.0000	1.7310	0.9980

**Table 4 sensors-19-02646-t004:** Values of the *p*-values for each gas sensor.

Sensor	*p*-Value
MQ-2	0.000000e+00
MQ-3	0.000000e+00
MQ-4	0.000000e+00
MQ-5	0.000000e+00
MQ-6	0.000000e+00
MQ-7	0.000000e+00
MQ-8	0.000000e+00
MQ-9	0.000000e+00
MQ-135	3.725089e-01
TGS822	9.494413e-02
TGS2600	0.000000e+00
TGS2602	0.000000e+00
TGS2603	0.000000e+00

**Table 5 sensors-19-02646-t005:** Methods and their parameters used.

Method	Parameters
MLR	R^2^ = 0.888 and R^2^adj = 0.888
ELM	*nhid* = 15 and *actfun = purelin*
RF	*ntree* = 500 and *mtry* = 4
MNLR	R^2^ = 0.811; R^2^adj = 0.811, and logarith transformation

**Table 6 sensors-19-02646-t006:** Predicted values and the RMSE of the test samples for each method.

Method	RMSE Validation (10-Fold)	RMSE Test
MLR	0.58	0.45
ELM	0.63	0.33
RF	0.74	1.19
MNLR	0.76	0.97

**Table 7 sensors-19-02646-t007:** Percent error of test samples for each method.

Method	C4	C4M	C4.5	C4.6	C5	C5.4	C7.9	Average
MLR	9.72%	1.38%	5.63%	8.88%	5.93%	14.48%	7.29%	7.62%
ELM	0.91%	4.54%	5.03%	7.33%	9.64%	10.78%	0.08%	5.47%
RF	3.76%	18.12%	44.42%	31.53%	35.39%	7.12%	0.271%	20.09%
MNLR	0.36%	6.92%	0.34%	8.34%	5.15%	18.47%	29.22%	9.83%

**Table 8 sensors-19-02646-t008:** Summary of results of related work on the classification of beverages.

Objective	Techniques Used	Parameters	References
Detection of alcohol content	RSM	R^2^ = 0.991	Nurul et al. [31]
Wine classification	PLS e ANN	R^2^ = 0.653 e R^2^ = 0.844	Aleixandre et al. [32]
Beer classification	PCA	Variance = 87.1%	Ragazzo-Sanchez et al. [33]
Beer classification	PCA e SVM	Error = 0% (Validation and test)	Ghasemi-Varnamkhasti et al. [34]
Detection of alcohol content of beers	ELM e MLR	ELM − Error = 5.47% and RMSE = 0.33 (Test); MLR − Error = 7.62%, RMSE = 0.45 and R^2^ = 0.888 (Test);	This work

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
