# Peer review of "A Prototype to Detect the Alcohol Content of Beers Based on an Electronic Nose"

_sensors, 2019, doi:10.3390/s19112646_

Reviewer 1 Report

I reviewed the paper entitled: "Prototype to detect the alcohol content of beers based on electronic nose" with the following remarks:

1) Sentences and grammar of the English language should be check

2) This study has some experimental issues that must be solve:

- How the authors controlled the pressure, temperature and humidity?? please, describes very well this subjects.

- Indicate ¿which was the sample rate of data acquisition card (i.e, samples/sec) used to make the differents test??

- In Figures 5, place the units (volts) for each signal

3) Place the rest of "p-value" for each sensor in a table to validate the Hypothesis

4) Figures and Tables should be describes (on the text) before the Table and Figure.

5) Table 6 illustrated the percent error of test samples. I think that these errors were very hight.

Would be necessary to apply a pattern recognition method to see how would be possible to improve these obtained results.

6) Some references should be verified

 Author Response

Dear reviewers,

Firstly, we would like to thank you for your attention and your contributions to our manuscript. We also appreciate the opportunity to resubmit the revised manuscript.

We try to answer every item of each reviewer and also do a new overhaul to minimize any problems with the language.

We are very much hopeful that the manuscript has now responded to the open questions and comments so it can be accepted for publication.

Regards,

Dr. Sergio Stevan Jr.

Reviewer 1

Question / appointment

Answers / comments

Sentences and grammar of the English language should be   check.

Dear reviewer, we   checked the manuscript again. We have corrected several minor flaws and tried   to improve the fluency of the manuscript. The main adjustments / changes were   in the lines: 57, 59, 101, 102, 106, 110, 135, 163, 184, 208, 236, 281, 282,   311, 318, 418,   423, 424, 428, 431, 435, 437, 439, 443, 444, 445, 446,   448, 449, 450, 452, and 453.

How the authors   controlled the pressure, temperature, and   humidity?? please, describes very well these subjects.

We improve this   information. This was described better in lines 149 and 228.

Indicate which was the sample rate of data acquisition card (i.e., samples/sec) used to   make the different tests.

Ok, it was added on   line 199.

In Figures 5, place the units (volts) for each signal.

The units were added   in the axes title in graphics of Figure 5.

Place the rest of     "p-value" for each sensor in a table to validate the   Hypothesis.

Dear reviewer,   according to his orientation, was inserted a new table (Table 4).

Figures and Tables should be described (on the text) before   the Table and Figure.

We insert description   after Table 1, Figure 5, Table 5, Figure 6, Table 7 and Table were placed   above.

Table 6 illustrated the percent error of the test   samples.   I think that these errors were very hight.

Would be necessary to apply a pattern recognition method to see   how would be possible to improve these   obtained results.

Dear reviewer, thanks   for your appointment.    For your better understanding, we added:
  -- The Line 242 to explain why the regression was chosen;
  As Table 7 shows the results obtained in this work and correlated works; we added:
  -- Lines 405 and 425 explain that the results   obtained were good in comparison with other works in the literature.

Some references should be   verified.

Dear reviewer, we   checked the references again and applied a tool to adjust them.

In addition to these points, several other improvements were made to provide better reading fluency and to correct minor misunderstandings. Thus, we are sure that your collaboration was very important for the paper to improve quality and thus be able to meet the premises of this important journal.
Thank you very much for your attention and review.

Reviewer 2 Report

I find your work interesting and clear in the data analysis. Only some suggestion to ameliorate the text clarity especially in the experimental part:

in row 57 is better to cancel "and they are these disadvantages".

in the row there is an R^2 of 0.052, maybe there's a zero too much?

Is not clear the e-nose how much measurements do. I understand that the training was made on 25 samples but before you write that the standards are 15. The training Dataset was composed by 12  experiment and in each experiment you measure all the 25 samples, is right? Beer types are 7 and the samples are 14 but there is no mention on the beer Dataset. I suppose that you measure the samples only in one experiment, is true?

In row 163 i understand that a measurement last 8 minute and sampling happen every second. So is better to use the term sampling instead of measurement when you write "each measurement lasted 1 second"

in row 191 I suppose that pi is the sample "i" of the measure p. Is better to declare it.

In the equation 1 you explicit y(Rs, x0, x1) but Rs doesn't appear in the equation. I understand that if i have an experimental value pi this must be corrected adding or multiplying the value y. Is true?

From the equation 2 i understand that p_inorm is included between 0 and 1 but in fig 5 you write of sensor response without define it. 

Author Response

Dear reviewers,

Firstly, we would like to thank you for your attention and your contributions to our manuscript. We also appreciate the opportunity to resubmit the revised manuscript.

We try to answer every item of each reviewer and also do a new overhaul to minimize any problems with the language.

We are very much hopeful that the manuscript has now responded to the open questions and comments so it can be accepted for publication.

Regards,

Dr. Sergio Stevan Jr.

Reviewer 2

Question / appointment

answers / comments

in row 57 is better to cancel "and they are these   disadvantages".

This sentence was removed.

in the row there is an R² of 0.052, maybe there's a zero too much?

We verify again.  The value is correct.

Is not clear the e-nose how much measurements do. I   understand that the training was made on   25 samples but before you   write that the standards are 15. The training   Dataset was composed of   12 experiments and in each experiment,   you measure all the 25 samples, is right? Beer types are 7 and the samples are 14 but there is no mention on the   beer Dataset. I   suppose that you measure the samples only in one   experiment, is true?

Thank you for your   important note. There was really a mistake and the amount was corrected. The   value of 25 was replaced by 15. Therefore, 6 experiments were performed and   each experiment had 15 samples. The test was performed with 7 different types   of beers, with 2 samples each. The measurement of each sample was performed   in one experiment.

In row 163 I   understand that measurement last 8 minute and sampling happen every second.   So is better to use the term sampling   instead of measurement when you write   "each measurement lasted 1   second".

Dear reviewer, to   improve this lecture, the term has   been replaced by “sampling”

in row 191 I suppose   that pi is the sample "i" of the measure p. Is better to declare it.

To clarify this   appointment, the description of “i” has been added.

In equation 1, you   explicit y(Rs, x0, x1) but Rs doesn't appear in the equation. I     understand that if I have an experimental value pi this must be corrected adding or multiplying the value y.   Is true?

Thanks for your   appointment. We verify that the equation was incomplete. To correct, we   insert the variable Rs into the numerator of the equation.

From equation 2 I   understand that p_inorm is included   between 0 and 1 but in fig 5 you write of sensor response without defining   it.

We prefer to show the results of the sensor responses before   applying the normalization. Therefore, Figure 5 was maintained.

In addition to these points, several other improvements were made to provide better reading fluency and to correct minor misunderstandings. Thus, we are sure that your collaboration was very important for the paper to improve quality and thus be able to meet the premises of this important journal.
Thank you very much for your attention and review.

Round  2

Reviewer 1 Report

Thanks to the authors for the corrections performed.

Please, Could you explain very well why you did not  apply some pattern recognition method (e.g., PCA or SMV) to compare the obtained results.

Author Response

Dear reviewer,

In answer to your question, I say: for our case, regression results in better results, and we include this clarification in the article (between lines 235 and 244). However, we have included here, only in response to the reviewer, the study regarding pattern recognition methods. Thank you in advance for your cooperation, and we hope to have succeeded in responding to it.

Thank you.
